# Assessment of the Variability of Many Years of Thunderstorm Activity in the Aspect of Potential Threats to Aircraft at Selected Airports in Poland

**DOI:** 10.3390/ijerph17010144

**Published:** 2019-12-24

**Authors:** Małgorzata Kirschenstein, Kamil Krasuski, Jaroslaw Kozuba, Miroslav Kelemen

**Affiliations:** 1Institute of Navigation, Military University of Aviation, 08521 Deblin, Poland; m.kirschenstein@law.mil.pl (M.K.); k.krasuski@law.mil.pl (K.K.); 2Faculty of Transport, Silesian University of Technology, 44100 Gliwice; Poland; 3Faculty of Aeronautics, Technical University of Kosice, 04121 Kosice, Slovakia; miroslav.kelemen@tuke.sk

**Keywords:** variability, frequency of storm phenomena, airports, Poland

## Abstract

The article presents an assessment of the long-term variability of storm activity in the aspect of potential threats to aircraft. The analysis of data from the period 1970–2018 was conducted for selected airports in Poland: Gdańsk Lech Wałęsa Airport, IATA code: GDN, ICAO code: EPGD (54°22′39″N 18°27′59″E, altitude above sea level 149 m above sea level); Solidarity Szczecin- Goleniow Airport, IATA code: SZZ, ICAO code: EPSC (53°35′05″ N 14°54′08″ E, altitude above sea level 47 m above sea level); Poznań-Ławica Henryk Wieniawski Airport, IATA code: POZ, ICAO code: EPPO (52°25′16″ N 16°49′35″ E, altitude above sea level 94 m above sea level); Warsaw Chopin Airport, IATA code: WAW, ICAO code: EPWA (52°09′57″ N 20°58′02″ E, altitude above sea level 110 m above sea level); Copernicus Airport Wrocław, IATA code: WRO, ICAO code: EPWR (51°06′10″ N 16°53′10″ E, altitude above sea level 123 m above sea level); John Paul II International Airport Kraków-Balice, IATA code: KRK, ICAO code: EPKK (50°04′40″ N 19°47′06″ E, altitude above sea level 241 m above sea level). The purpose of this paper is to assess the long-term variability of storm activity in the aspect of potential threats to air operations in Poland with the examples of six selected airports. In order to achieve the goal, an analysis of the frequency of storm phenomena in Poland was carried out both in annual and long- term terms. The analysis will allow the assessment of the geographical diversity of the distribution of storm phenomena and their variability in the years 1970–2018. The next stage of the work will be to determine the climatic conditions that exert the greatest impact on the formation of storms. The important factors include atmospheric circulation, which, over the Polish territory, is shaped by the influence of air masses from the Atlantic Ocean, the Baltic Sea and in addition, from the vast continental area. All these air masses clash over the area of Poland causing large variability in the frequency of occurrence of hazardous atmospheric phenomena. For this reason, the Polish climate is defined as a moderate warm climate with transitory features. The important factors affecting regional diversity are local conditions, such as terrain, nature of the land, and distance from water reservoirs. The thermal, humidity and aerodynamic properties of the substrate, which are components of radiation processes, determine the exchange of energy at the interface between the atmosphere and the earth, and largely determine the intensity of selected hazardous atmospheric phenomena. Each occurrence of a storm is a potentially dangerous meteorological event that threatens the environment and human activities, including all types of transport. The studied phenomenon of storms is particularly dangerous for air transport. Literature shows that storm phenomena in Poland are characterized by a large regional diversity, both during the year and over many years. The greatest threat of storm phenomena occurs in the warm period of the year—spring and summer.

## 1. Introduction

A progressive global climate change, caused by both natural and anthropogenic factors, will increasingly affect the lives of the population and the world of fauna and flora. According to the IPCC report (2019), the water level will increase drastically and there will be more and more storms that will also be observed in Poland. The air temperature will increase (2.5–4.5 °C). The summers will be hotter, 35-degree heat will be more frequent. At such a high temperature, there will be storms, accompanied more often by hail and tornadoes. The weather will become extremely changeable. Winters will get warmer and it will snow less frequently. There will be more rainfall, and it will be more intense than of today. Drought will occur more often in the warm half-year. The second half of autumn, winter, and the beginning of spring are the periods when precipitation will be more frequent in the future.

In climate change studies, apart from basic meteorological elements—air temperature and atmospheric precipitation—analyses of dangerous atmospheric phenomena, which include the storm and accompanying atmospheric phenomena, also play an important role. In many cases, the course of these phenomena can be determined as an extreme meteorological event. A storm phenomenon is a set of simultaneous accompanying phenomena, such as atmospheric discharges, intense precipitation, including hail, turbulence, squall, fault, gusts, tornado, icing, as well as changes in atmospheric pressure and air temperature. Storms pose a threat to aircraft. This knowledge is necessary when planning and performing flights, both by pilots and Unmanned Aerial Vehicles operators. 

Therefore, the negative effects that these phenomena can cause are the reason for addressing this problem in this paper. The analysis of the frequency of storm phenomena carried out in the article not only during the year but also in the aspect of long-term changes will enable the authors to assess whether there are visible changes in the severity of dangerous meteorological phenomena, resulting from global climate changes.

## 2. Related Papers 

In Poland, the first storm studies concerned the time variability of the occurrence of storms and the determination of storm regions [1,2,3,4,5]. The studies also drew attention to changes in long- term storms, an increase in their frequencies in the cold half-year, and their relationship with synoptic situations [6,7]. The changing tendencies were examined. Also, a storm activity was forecast in Poland [8]. Storm precipitation was described to a much lesser extent. The last years of the study of hailstorms focused on their long-term variability, the diurnal course, and the circulatory conditions conducive to their occurrence [6,9]. 

It should be emphasized that a storm is a phenomenon that occurs frequently on a global scale. Storms can form from equatorial to polar zones. In the tropic latitudes, thunderstorms appear throughout the year. In moderate latitudes, storms most often occur in spring, summer, and autumn. On a global scale, the average annual number of storm days varies, from more than 160 storms in the equatorial zone to around 20–30 in moderate latitudes. In Poland, according to Bielec-Bąkowska [6], the average annual number of stormy days varies from 15 days on the coast to 33 days in the Tatras and in the south-eastern regions of the country. Approximately, 97% of all storms occur from April to September. With the increase of latitude due to lower temperatures, the vertical range of atmospheric storms decreases, and altitude affects the potential intensity of the storm. The varied inflow of solar radiation in the course of the year affects the distribution of air temperature and the intensity of convective movements. 

There are also publications on storm phenomena on a regional scale. The development of storm phenomena for the north-west area of Poland is also worth stressing. It has been shown that due to the prevailing circulation and thermal conditions, this area differs from other regions of the country. In this area, air masses from the ocean, sea, and continent clash with one another. This leads to great variability of weather conditions. The coast is often within the range of cyclones that travel from the North Atlantic towards northeastern Europe. As a result, larger masses of the ocean air participate in the overall atmospheric circulation. This affects the formation of storm phenomena [2,10,11,12]. There are also publications on dangerous atmospheric phenomena, including storms for the selected airports [13,14,15].

Because each occurrence of a storm should be treated as a potentially dangerous phenomenon, the work investigated which factors affect the frequency and distribution of storms in Poland. The linear regression trend was also calculated, determining the variability of the occurrence of storms in the period 1970–2018.

## 3. Research Method

Studies that show changes in a given meteorological element in different regions and which have been developed for long measurement series are vitally important. With such a comprehensive analysis, it is possible to demonstrate the interdependence of meteorological elements, regional differentiation, and determine which set of climate-forming factors and processes has the greatest impact on their variability. As a result, the characteristics of the climate selected for the study of the area will be better understood. We will not manage to obtain a full picture of the climate features by analyzing the individual elements or individual stations where climate conditions are largely determined by local factors.

In relation to the above, an attempt was made to determine the variability of storm phenomena at selected airports in Poland in the years 1970–2018. The study will contain an analysis of storm phenomena, not only in annual and multiannual sections, but also in a spatial distribution. Storm frequency strings will be examined to detect variability in their course. Six airports located in different parts of Poland were selected for the analysis. The development of the storms was based on meteorological data from the Polish Institute of Meteorology and Water Management. The data cover the 49-year period 1970–2018 for airports. Meteorological measurements and observations are carried out systematically, using the basic networks of stations and posts as well as special measuring networks. The data are reliable and homogeneous. The measurements are conducted in accordance with WMO (World Meteorological Organization) standards.

Statistical data for each airport were compiled for each airport: average monthly and annual number of days with a storm, maximum value and minimum number of days with a storm, average number of days with a storm for the seasons, average monthly and annual duration of the storm in hours, average monthly and annual duration of one storm over the airport. The spatial distributions were made in the SURFER15 graphic program.

In order to assess changes in the frequency of storm phenomena as a function of time, a linear regression method was used, which determines an increase or a decrease in the frequency of storms per unit of time. The linear trend is a special case of linear regression where the explanatory variable X is the time variable t. In this case we say that we deal with a time series, i.e., data that are ordered by time.

The definition of the linear trend function is expressed by the formula
*y = a⋅t + b*
*a—*trend directional factor*; b—*free expression of the trend, *t—*values of variables When a > 0 we deal with a growing trend. The larger the “a” value, the faster the Y value increases over time. When a < 0 we deal with a decreasing trend. The smaller the “a” value, the faster the Y value decreases over time. 

## 4. Research Test and Results

Storms are associated with the occurrence of the Cumulonimbus cloud (Cb—storm cloud). Cumulonimbus clouds are dense clouds with a significant vertical extent. The lower base is on average at 300–600 m, while the upper limit of clouds exceeds 14,000 m (depending on the thickness of the troposphere). Cumulonimbus occurs as a separate cell or forms a series of clouds. The cloud gives a fleeting rainfall, usually of high intensity and stormy nature, with a simultaneous occurrence of violent gusts of wind. 

The definition employed by IMGW stations (the Polish Institute of Meteorology and Water Management) and binding since 1962 is used to describe the storm. The storm is considered to be a *“thunder heard after less than 10 s from the time of seeing the lightning, and the distant storm is called the phenomenon of a thunder occurring after more than 10 s, possibly accompanied by a lightning. The beginning of the storm marks the moment of hearing the first thunder, while the end is a thunder, after which within 15 min, the next one cannot be heard”* [3].

Storms are created as a result of a rapid rise of humid and warm air to higher altitudes—a thermal storm, or a strong collision of humid and warm air with a mass of cool air—a front storm [12]. 

Storms can also form on the windward side of the mountains when the warm and humid air flows—the orographic storm. North Poland is an area, in which storms occur less frequently than in the rest of Poland. The studied area is varied in terms of orography. The elevations in the Pomeranian Lake District and the influence of the Baltic Sea make the frequency of storms highly variable, both in the course of one or many years. 

The analysis shows that the average annual number of days with a storm (of six airports) for the entire area is 23. In the multiannual 1970–2018, fluctuations in the average annual number of days with a storm were very large. At airports they were as follows: EPGD—from 5 to 28 storms; EPSC—from 7 to 30 storms; EPPO—from 12 to 36 storms; EPWA—from 13 to 34 storms; EPWR—from 15 to 37 storms; EPKK—from 17 to 39 storms. Consequently, the amplitude ranged from 21 (EPWA) to 24 storms (EPPO), (see Figure 1, Figure 2, and Figure 3). 

There were big differences in the geographical distribution—storms were less frequent in the northern part of the studied area and in the Baltic coast area than in southern Poland. The average annual number of days with a storm ranged from 17 in EPGD to 29 in EPKK (see Figure 4). The maximum average annual number of days with a storm ranged from 28 in EPGD to 39 in EPKK.

A data analysis shows that storms most often occurred in the warm months of the year. They appeared very rarely in the winter months. However, most often storms occurred in the summer months, with a maximum in July. In July, the frequency of storms ranged from about 4 storms in EPGD to 7 storms in EPKK. In the years 1970–2018, the most storms—15 (in 2012)—were recorded in EPPO (see Table 1). A frequent occurrence of storms in the summer months is caused by high air temperatures, intense evaporation, which promote the development of intensive convection and the formation of cumulonimbus clouds. In addition, in the warm season of the year, the influx of cool air masses from higher latitudes or from above water bodies over a warm land area also promotes the development of convection and the formation of storm clouds.

When analyzing the frequency of storms in the seasons, it was found that the most days with a storm occurred in the summer—the frequency ranged from 62.9% (EPSC) to 67.3% (EPPO). Often, storms also occurred in spring, and their incidence ranged from 22.6% (EPGD) to 26.3% (EPKK). Occasionally storms appeared in autumn—from 6.8% (EPPO, EPWR) to 9.9% (EPGD). In the winter, however, due to low temperatures storms were rare and occurred with a frequency of 0.6% (EPGD) to 2.0% (EPSC). The analysis shows (see Table 1) that in the spring and summer, the most favorable conditions for creating storms occurred in southern Poland, while the least frequently near the Baltic coast. In the autumn, however, the increase in the frequency of storms is clearly influenced by warmer and humid air masses coming from the Baltic Sea. In winter, storms occurred sporadically at every airport.

It should be emphasized that in the whole period (1970–2018), there were significant differences in the studied area in individual seasons. The average number of stormy days for the studied area Poland was: spring—6, summer—15, autumn—2 and winter around 0.3. Over the years, the differences were large. In the spring, although storms appeared at all airports with a similar frequency, from about 4 to 8 storms, in the course of many years their number reached even 14 storms (in EPWA and EPPO). In the summer, apart from frontal storms, intra-mass storms often occur. Overall, this is the season of the year characterized by both the largest number of days with a storm (on average 11–19 storms) and the largest spatial diversity. In the summer, up to 30 storms were recorded at EPKK airport, while in EPSC a maximum of 21 storms occurred (see Table 1).

In aviation, the duration of the storm is important information. The analysis of the data shows that, on average, storms lasted about 35.5 h a year. At individual airports, the differences were large and ranged from 22.4 (EPGD) to 53.2 h (EPKK), (see Figure 5). In the multiannual course, the maximum average annual values ranged from 41.2 (EPGD) to 92.5 h (EPKK). In the annual course, the longest storms lasted in the summer months, with a maximum in July: from 5.9 h (in EPSC) to 13.4 h (in EPKK). In the years 1970–2018, the maximum duration of one storm was recorded in EPKK—41.7 h in 1971 (see Table 2). 

The analysis also took into account the average duration of one storm. The analysis shows that one medium storm lasted from 1.3 h (EPSC) to 1.9 h (EPKK). When analyzing the duration of one storm, it was found that the longest storms also occurred in the summer months. In EPKK, a single storm lasted up to 2 h (see Table 3).

The assessment of storm frequency variability in time was based on a linear regression analysis that determines the increase or decrease in storm frequency per unit of time. The analysis of the linear trend coefficients showed that in the years 1970–2018, in the studied area of Poland, the linear trend was positive (except for a slight decrease in EPWR) for the average annual number of days with a storm. The spatial distribution shows that the average annual number of days with a storm showed the largest upward trend of 1.5 days/10 years in the north—especially at EPGD. In contrast, the frequency of storms did not change at EPWR airport. At the EPGD and EPPO airports, the trend of increasing the frequency of storms meant that the average duration of storms in the year increased (see Figure 6). At other airports, despite the tendency to increase the frequency of storms, there was a tendency to decrease their duration. The largest decrease was observed in EPKK—3 h/10 years (see Figure 7). In addition, it was observed that at all airports individual storms tend to be shorter. 

The above analysis indicates that storms occur more frequently, yet they are increasingly shorter. This means that their dynamics is greater. This is due to a general increase in air temperature. As a result of higher air temperatures, the intensity of convective movements increases. These movements lead to a rise in the vertical range of atmospheric storms, whereas the height affects their potential intensity. 

The analysis showed that the air temperature influences the formation of storms. Therefore, the article also presents an analysis of changes in air temperature. The analysis included data from the Polish Institute of Meteorology and Water Management from the period 1970–2018 for the airports: EPGD, EPSC, EPPO, EPWA, EPWR, EPKK. It was examined that the average annual air temperature in the years 1970–2018 ranged from 8.0 °C at EPGD airport to 9.1 °C at EPWR airport (see Figure 8).

The analysis of statistical data of the linear trend confirms the increase in air temperature (see Figure 9, Figure 10 and Figure 11).

In the years 1970–2018, in all the selected airports, there was an increase in air temperature, as follows: 

linear trend y = 0.0351 °C/year (an increase in air temperature over a period of 49 years by approximately 1.7 °C) in EPSC;

linear trend y = 0.0344/year (an increase in air temperature over a period of 49 years by approximately 1.7 °C) in EPGD;

linear trend y = 0.0418 °C/year (an increase in air temperature over a period of 49 years by approximately 2.0 °C) in EPPO;

linear trend y = 0.0434 °C/year (an increase in air temperature over a period of 49 years by approximately 2.1 °C) in EPWA;

linear trend y = 0.0491 °C/year (an increase in air temperature over a period of 49 years by approximately 2.4 °C) in EPWR;

linear trend y = 0.0402 °C/year (an increase in air temperature over a period of 49 years by approximately 2.0 °C) in EPKK. 

## 5. Discussion

Monitoring thunderstorms and related atmospheric phenomena play a very important role in supervising the movement of aircraft. Tracking storm activity allows an effective avoidance of threats and facilitates safe air traffic. Threats to aircraft associated with the occurrence of storms include lightning, ice, hail and heavy rainfall, intense turbulence, squall, wind shear, wind gusts, and tornado. 

Lightning is a direct threat—it is a danger to aircraft in flight due to the possibility of a direct lightning strike, and therefore the possibility of damage to the aircraft, damage to electronics, and large altimeter errors may appear. Precipitation creates an electrostatic field that causes interference at all radio frequencies (short, medium, and long waves). 

In addition, an aircraft that is in the area of a storm cloud can itself become an object that releases a lightning strike, because its surface, and especially its pointed elements, are the source of corona discharge. If a storm cannot be bypassed, the pilot uses procedures to describe flying through the storm to minimize the risk of a disaster. According to the procedure, in the event of high variability and dynamics of a storm phenomenon, the pilot must decide to correct the heading and altitude. 

When making decisions, the pilot should take into account current messages and be guided by the following principles:

Do not fly into the storm if the storm covers 6/10 or more of the area.

Do not approach a storm less than 30 km if the storm is identified as strong (gives a strong echo on the radar).

In a situation where the pilot must fly into a storm, then before entering they must check the on- board instruments.

They should try to maintain a constant flight altitude.

They should choose the heading that will allow the shortest flight during a storm.

If the flight must take place in a storm, do not turn back. This maneuver will only make the pilot remain in the storm zone even longer, and in addition the maneuver will increase the loads on the aircraft and increase the risk of stalling, and may result in the pilot losing spatial orientation or losing too much height. 

However, a military pilot in accordance with Flight Regulations uses the following procedure: “Flying in strongly uplifted clouds and approaching storm clouds at a horizontal distance of less than 10 km is prohibited. It is also prohibited to fly under strongly uplifted clouds that give heavy rain, hail, snow or lightning. It is permitted to fly over encountered storm clouds at no less than 1000 m. 

Depending on the altitude and phase of the flight, the discharges can be cloud or ground. The moments of takeoff and landing of the aircraft in the presence of storm clouds are particularly dangerous. The impact of atmospheric discharges on ground air traffic services is also a threat. A particular danger may arise during lightning at the time of, for example, refueling. In addition, the presence of a storm can cause delays at a given airport, and thus disorganization of air traffic over a large area.

From the point of view of meteorological protection of air traffic, in addition to observing atmospheric discharges, it is also very important to analyze other meteorological phenomena related to the storm, such as: the occurrence of intense turbulence, icing, strong ascending and descending currents, and intense rainfall, or tornadoes. These phenomena have a great impact on the safe flight of an aircraft. Therefore, information on the activity of these phenomena is a valuable source of information used by aircraft crews and supplemented with other data, for example radar and satellite.

In view of the above, it appears that the study of storm phenomena is crucial and constitutes important information for air operations. The obtained results confirm the fact that global warming contributes to an increase in the frequency of storm phenomena. This is due to a general increase in air temperature. As a result of higher air temperatures, the intensity of evaporation and convective movements increases. These movements increase the vertical range of atmospheric storms, and the altitude affects their potential intensity. In addition, research shows that storms are becoming more frequent, but their duration is becoming shorter. This means that their dynamics is greater, which increases the threat to aircraft.

It should be emphasized that dangerous atmospheric phenomena are one of the important elements determining the safe performance of aerial operations. Statistics show that 630 air disasters caused by dangerous atmospheric phenomena took place in the world in the years 1945–2018: storms (26), lightning strike (29), sandstorm (2), turbulence/side wind (91), wind shear (116), icing (223), rainfall (9), low visibility (132), volcanic ash (2) [16].

## 6. Conclusions

Each occurrence of a storm is a potentially dangerous meteorological event threatening the safe performance of flight operations. An analysis of these phenomena at selected airports in Poland indicates a large regional diversity, both in an annual and long-term course. The average annual number of storm days for the whole area equals 23. Maximum storms occurred in July. There were big differences in the geographical distribution—storms were less frequent in the western part of the studied area and in the Baltic coast zone than in the south of Poland. The average annual number of days with a storm ranged from 17 to 29. At all airports their maximum frequency reached 28–39 storms a year, but there were also years in which storms occasionally appeared 5–17. Very large fluctuations occurred in the summer months. In the annual course of 88.8%–92.4% of storms occurred in spring and summer. It was found that in the years 1970–2018, the geographical distribution of storm frequency varied greatly. The gradient of changes was approximately north–south. The regional diversity is associated with the synoptic situations prevailing in a given area, which are largely modified by the impact of local geographical environmental conditions. This is particularly evident in the annual frequency of studied phenomena. Comparing the studied area, the impact of warm and moist air masses from the Baltic Sea and the Atlantic Ocean is clearly marked, which affects the extension of the storms period in the north of the country—in the autumn and even winter months. In turn, especially in the springtime, cool masses of air mean that storms appear less frequently. In the south of Poland, the orography of the area and the amount of solar energy supply, which is conducive to the formation of storms, especially in the spring and summer, exert a clear impact.

An analysis of the linear trend coefficients showed that in the years 1970–2018, at most airports (except EPWR), the average annual frequency of storms rose. At the EPGD and EPPO airports, the trend to increase the frequency of storms meant that the average duration of storms during the year increased. At other airports, despite the tendency to increase the frequency of storms, their duration decreased. The largest decrease was observed in EPKK. In addition, it has been observed that at all airports individual storms are becoming shorter and shorter. The above analysis shows that there will be an intensification of storm phenomena that may result in increased threats to flight operations. 

The results obtained confirm that global climate change is intensifying dangerous atmospheric phenomena, including storm phenomena. In the future, threats to flight operations will increase. In view of the above, there is a justification for extending the research towards analyzing data on hazardous atmospheric phenomena that may occur during a storm. These dangerous phenomena include hail, heavy rain and snow, icing, turbulence, wind faults, squalls and wind gusts. Their analysis will be the subject of research in subsequent works.

## Figures and Tables

**Figure 1 ijerph-17-00144-f001:**
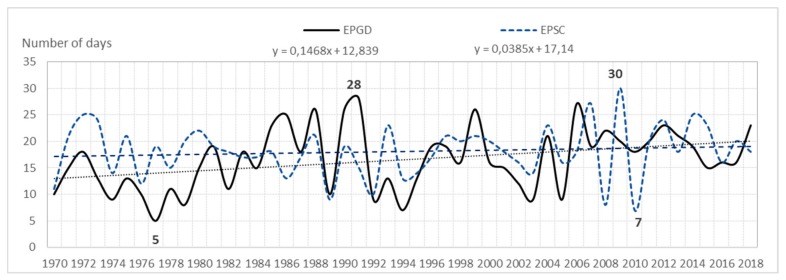
Average annual number of storm days and a linear trend in EPGD and EPSC (1970–2018).

**Figure 2 ijerph-17-00144-f002:**
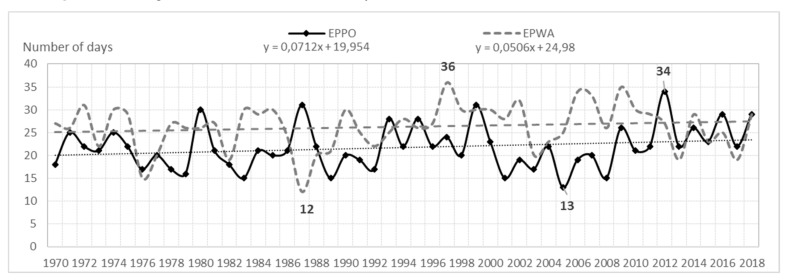
Average annual number of storm days and a linear trend in EPPO and EPWA (1970–2018).

**Figure 3 ijerph-17-00144-f003:**
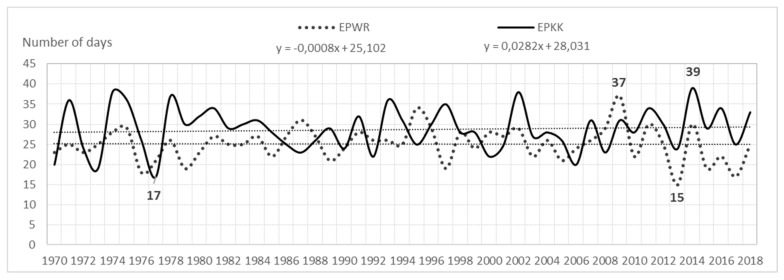
Average annual number of storm days and a linear trend in EPWR and EPKK (1970–2018).

**Figure 4 ijerph-17-00144-f004:**
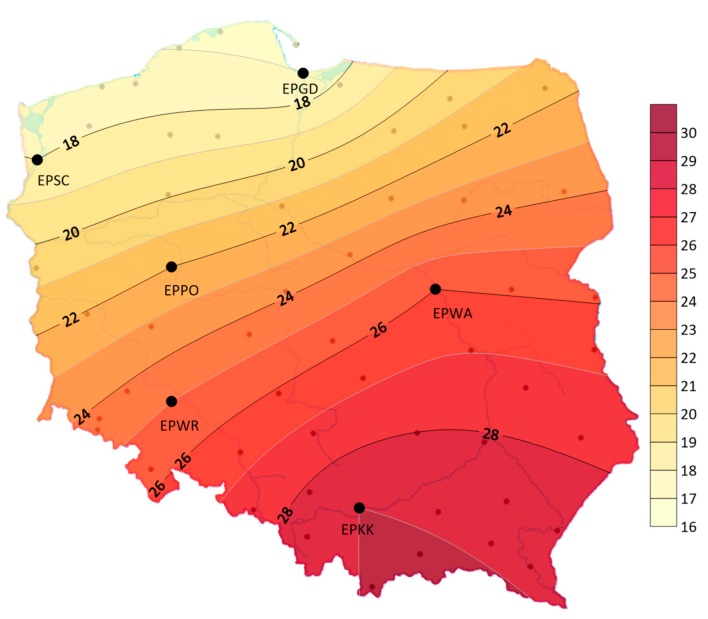
Average annual number of days with a storm in Poland (1970–2018).

**Figure 5 ijerph-17-00144-f005:**
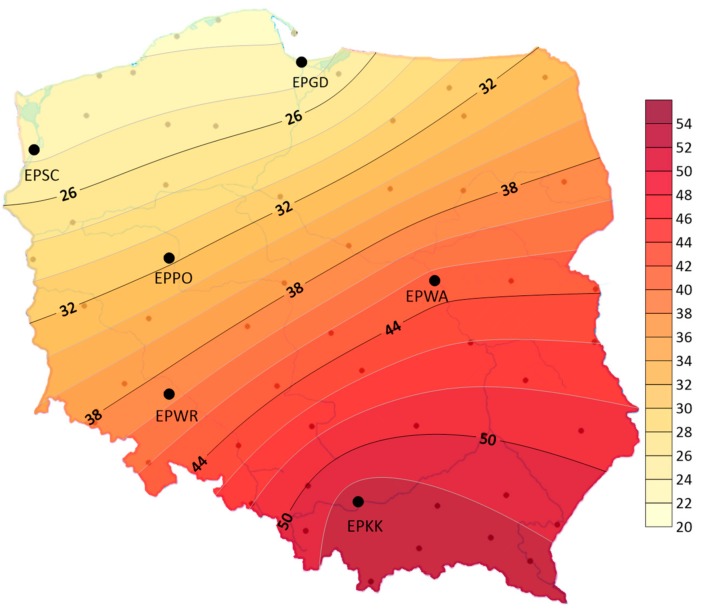
The average annual duration of storms in hours (1970–2018).

**Figure 6 ijerph-17-00144-f006:**
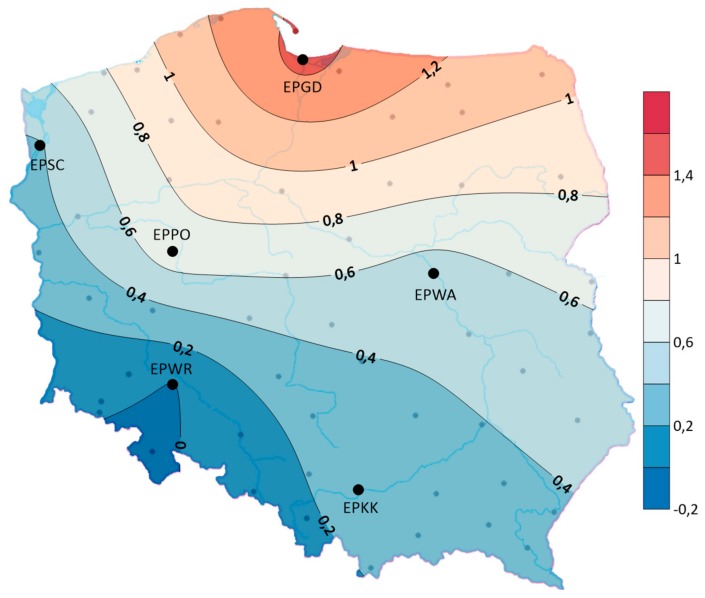
The distribution of the linear trend indicator for the average annual number of days with a storm (days/10 years), (1970–2018).

**Figure 7 ijerph-17-00144-f007:**
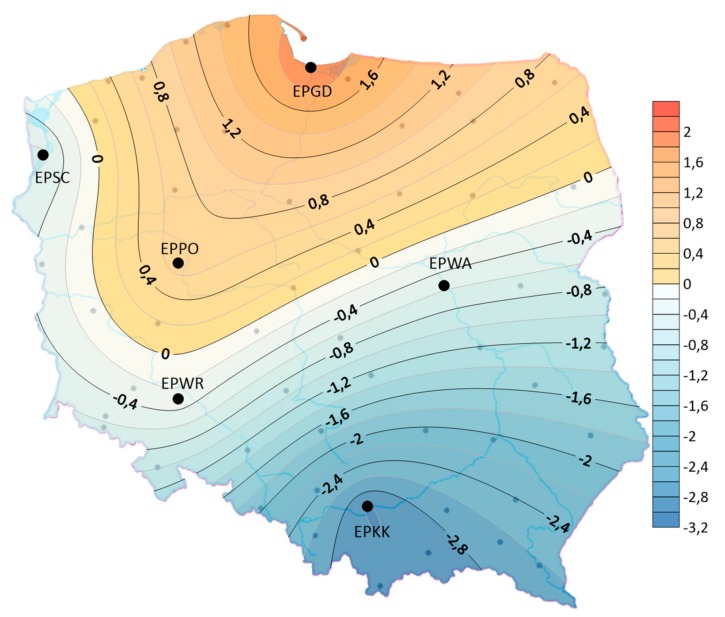
The distribution of the linear trend indicator of the average annual duration of the storm (hours/10 years), (1970–2018).

**Figure 8 ijerph-17-00144-f008:**
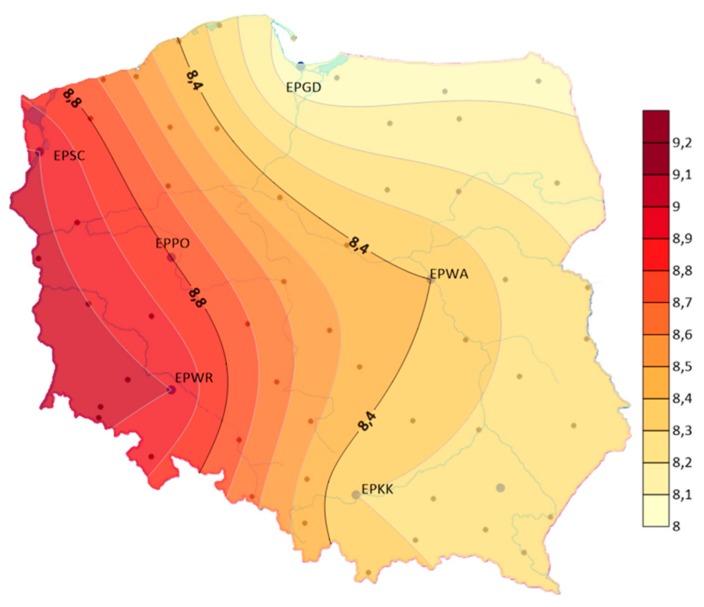
Average annual air temperature in the years 1970–2018.

**Figure 9 ijerph-17-00144-f009:**
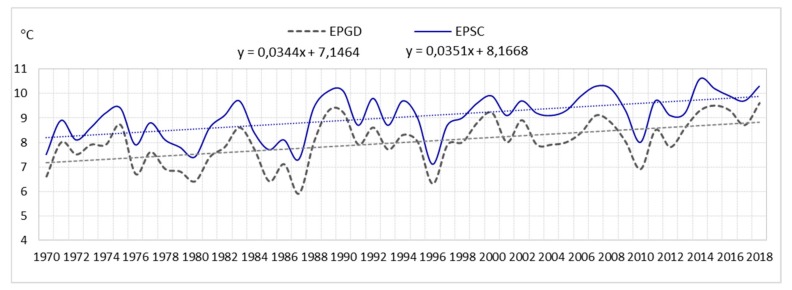
Average annual air temperature and linear trend in EPGD and EPSC (1970–2018).

**Figure 10 ijerph-17-00144-f010:**
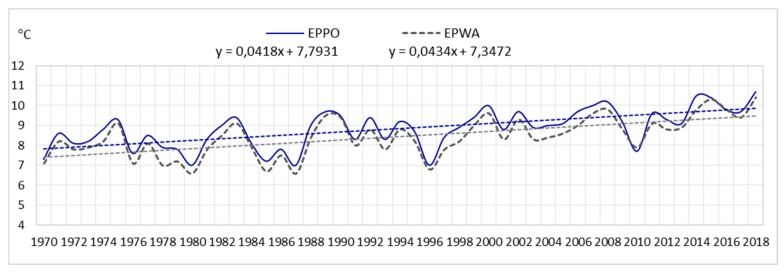
Average annual air temperature and linear trend in EPPO and EPWA (1970–2018).

**Figure 11 ijerph-17-00144-f011:**
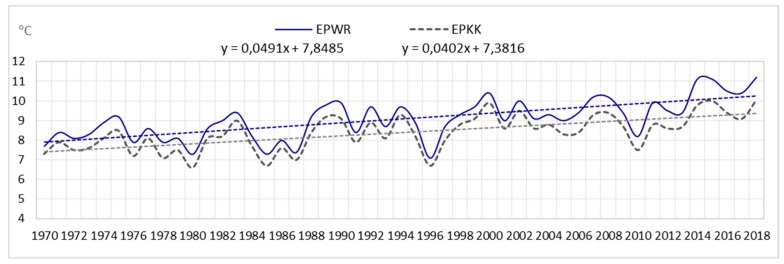
Average annual air temperature and linear trend in EPWR and EPKK (1970–2018).

**Table 1 ijerph-17-00144-t001:** This is. Average (Aver.) and maximum (MAX) number of days with a storm in Poland (1970–2018).

STATION	INDEX	I	II	III	IV	V	VI	VII	VIII	IX	X	XI	XII	YEAR	SPRING	SUMMER	AUTUMN	WINTER
**EPGD**	**Aver.**	**0.0**	**0.0**	**0.1**	**0.8**	**2.9**	**3.7**	**3.9**	**3.4**	**1.2**	**0.3**	**0.1**	**0.0**	**17**	**4**	**11**	**2**	**0.1**
	MAX	1	1	1	4	8	11	9	10	6	2	1	1	28	9	23	7	1
**EPSC**	**Aver.**	**0.1**	**0.1**	**0.2**	**0.8**	**3.7**	**4.0**	**4.2**	**3.2**	**1.3**	**0.2**	**0.1**	**0.1**	**18**	**5**	**11**	**2**	**0.4**
	MAX	2	2	2	4	9	10	10	8	5	2	1	1	30	12	21	6	2
**EPPO**	**Aver.**	**0.0**	**0.1**	**0.3**	**1.1**	**4.0**	**4.7**	**5.6**	**4.3**	**1.1**	**0.3**	**0.1**	**0.0**	**22**	**5**	**15**	**1**	**0.2**
	MAX	1	1	2	8	7	10	15	9	5	2	1	1	34	14	24	5	2
**EPWA**	**Aver.**	**0.1**	**0.2**	**0.5**	**1.6**	**4.6**	**5.7**	**6.6**	**5.0**	**1.3**	**0.4**	**0.2**	**0.1**	**26**	**7**	**17**	**2**	**0.3**
	MAX	1	2	3	5	11	10	12	12	6	2	2	2	36	14	29	7	2
**EPWR**	**Aver.**	**0.1**	**0.1**	**0.3**	**1.2**	**4.9**	**5.9**	**6.0**	**4.9**	**1.4**	**0.3**	**0.0**	**0.1**	**25**	**6**	**17**	**2**	**0.2**
	MAX	1	2	2	5	11	11	11	10	5	3	1	1	37	12	28	5	2
**EPKK**	**Aver.**	**0.2**	**0.1**	**0.4**	**1.6**	**5.6**	**6.7**	**6.9**	**5.2**	**1.7**	**0.2**	**0.1**	**0.1**	**29**	**8**	**19**	**2**	**0.3**
	MAX	2	2	3	7	11	13	12	11	5	2	1	1	39	13	30	6	2

Notes: **EPGD**—Gdańsk Lech Wałęsa Airport, **EPSC**—Solidarity Szczecin- Goleniow Airport, **EPPO**—Poznań-Ławica Henryk Wieniawski Airport, **EPWA**—Warsaw Chopin Airport, **EPWR**—Copernicus Airport Wrocław, **EPKK**—John Paul II International Airport Kraków-Balice.

**Table 2 ijerph-17-00144-t002:** Average (Aver.) and maximum annual (MAX) storm duration in hours (1970–2018).

STATION	INDEX	I	II	III	IV	V	VI	VII	VIII	IX	X	XI	XII	YEAR
**Average Annual (Aver.) and Maximum Annual (MAX) Storm Duration in Hours**														
**EPGD**	**Aver.**	**0.0**	**0.0**	**0.0**	**0.8**	**4.6**	**4.7**	**6.2**	**4.4**	**1.3**	**0.2**	**0.0**	**0.0**	**22.4**
	MAX	0.8	1.0	0.8	6.8	16.6	16.1	24.9	14.7	7.0	2.4	0.3	0.5	41.2
**EPSC**	**Aver.**	**0.0**	**0.0**	**0.2**	**0.7**	**5.2**	**5.0**	**5.9**	**4.7**	**1.3**	**0.2**	**0.1**	**0.1**	**23.4**
	MAX	0.9	0.6	2.0	6.7	21.4	14.8	19.7	16.3	7.1	2.3	1.0	0.8	42.8
**EPPO**	**Aver.**	**0.0**	**0.0**	**0.1**	**1.3**	**6.0**	**7.6**	**8.6**	**6.3**	**1.3**	**0.2**	**0.0**	**0.0**	**31.5**
	MAX	0.8	0.5	2.0	11.0	15.0	18.2	25.4	19.4	7.0	2.0	1.2	0.4	57.8
**EPWA**	**Aver.**	**0.1**	**0.1**	**0.3**	**1.8**	**7.9**	**10.6**	**11.5**	**8.3**	**1.7**	**0.4**	**0.1**	**0.1**	**42.8**
	MAX	1.1	2.5	2.2	7.6	27.9	32.0	31.0	19.8	8.2	5.7	1.0	2.3	73.9
**EPWR**	**Aver.**	**0.1**	**0.0**	**0.2**	**1.5**	**7.6**	**9.5**	**10.4**	**8.2**	**1.6**	**0.3**	**0.0**	**0.0**	**39.4**
	MAX	1.5	0.7	2.4	11.6	21.3	24.6	30.1	22.6	8.7	4.3	0.4	0.8	67.2
**EPKK**	**Aver.**	**0.2**	**0.0**	**0.4**	**2.1**	**11.2**	**13.4**	**13.4**	**9.7**	**2.6**	**0.3**	**0.0**	**0.0**	**53.2**
	MAX	1.7	0.9	2.3	13.8	26.5	41.7	29.3	22.6	11.2	4.9	0.7	1.0	92.3

**Table 3 ijerph-17-00144-t003:** Average (Aver.) duration of one storm in hours (1970–2018).

STATION	INDEX	I	II	III	IV	V	VI	VII	VIII	IX	X	XI	XII	YEAR
**Average Duration of One Storm in Hours**														
**EPGD**	Aver.	0.7	0.9	0.5	1.1	1.6	1.3	1.6	1.3	1.0	0.7	0.2	0.6	**1.4**
**EPSC**	Aver.	0.4	0.4	0.6	0.9	1.4	1.3	1.4	1.4	1.0	0.7	0.6	0.5	**1.3**
**EPPO**	Aver.	0.6	0.3	0.5	1.2	1.5	1.6	1.5	1.5	1.2	0.7	0.8	0.4	**1.5**
**EPWA**	Aver.	0.6	0.7	0.6	1.2	1.7	1.8	1.8	1.7	1.2	0.9	0.4	0.9	**1.6**
**EPWR**	Aver.	0.7	0.3	0.6	1.3	1.6	1.6	1.8	1.7	1.1	1.3	0.3	0.4	**1.6**
**EPKK**	Aver.	0.7	0.5	0.9	1.4	2.0	2.0	1.9	1.9	1.5	1.1	0.5	0.7	**1.9**

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
