# Peer review of "Assessment of the Variability of Many Years of Thunderstorm Activity in the Aspect of Potential Threats to Aircraft at Selected Airports in Poland"

_ijerph, 2019, doi:10.3390/ijerph17010144_

Round 1

Reviewer 1 Report

The article presents an assessment of changes in the frequency of storm phenomena at selected airports in Poland. The presented results confirm that global warming contributes to an increase in the frequency of storm activity, which may result in increased threats to flight operations.

Overally, I recommend accepting this paper after the following issues are addressed:
1. The first sentence in line 168 should be rewritten ("During the yearly storms could occur every month.")
2. The descriptions of Figure 6. and Figure 7. should be fully translated into English.

Author Response

Dear Reviewer,

Yours sincerely,

Jarosław KOZUBA, prof. SUT

Reviewer 2 Report

This paper presents a long term analysis of storm activity at 6 airports in Poland.  The paper concludes with a very important point, that while storm duration is decreasing, storm intensity is increasing at these locations, which presents increased risk to aircraft safety.  The use of English needs significant attention – as it is currently written it is difficult to understand parts of the paper.  Some examples are provided below, but these are just a few representations and are by no means exhaustive.

The layout of the paper is also a little odd, for example on page 3 line 114 statistics are introduced, same page line 119 a linear regression method is introduced, but these are not then followed by data.  The abstract seems to jump around, makes the same point multiple times, and presents future work in the middle.  The discussion also doesn’t seem to be entirely relevant to the data presented before it.

My major comments are that while the paper presents results worthy of publication, the science supporting said results needs to be improved.  Where statistics are presented, I feel that both the presentation could be improved, and calculating standard deviations or other appropriate statistical tests would greatly enhance the authors points.  All methods used need to be clearly defined, such that this study can be repeated, and thus contrasted, at other locations.  For example the entire paper revolves around storm activity, but it is not clear to me how the authors are defining “storms”.  The link to the change in storm characteristics to climate change, while almost certainly true, needs to be supported.

While I think the results are sound and worthy of publication, I encourage the authors to improve their science and presentation.

Specific comments:

Page 2 line 55 – the sentence beginning “In many cases, the course of these…..” does not make sense

Page 3 line 110 – you state the development of storms was based on meteorological data, please define the method used.  What parameters did you use to define “storm”.

Page 3 line 139 – is this the definition you used to define storm onset and duration?  Are these subjective data completely reliable?

Page 3 line 147 – it would be helpful to the reader to include a map, showing relevant orography and the locations of the airports in the study.  At a minimum, please refer the reader to figure 4.

Page 3 line 152 – do these numbers refer to the minimum and maximum number of storms occurring at each site between the years 1970 and 2018?  The figures indicate it is the annual average for each site?  Please make this point clear.

Page 5 line 181 – does the change in storm locations with seasons follow the prevailing wind patterns?

Page 5 line 187 – it is not clear what “the studied area” refers to?

Page 6 paragraph beginning line 199 – when you say duration, do you mean there was a storm present over the airport sites?  As per the comment for page 3 line 10, please clearly define your criteria.  Also, is there any data available on storms which may have passed over multiple airport sites?

Page 6 line 206 and throughout – I think you are using “one storm” where you mean “storms”.

Page 8 line 239 – can you say for certain your results are related to an increase in air temperature?  At a minimum please include a plot for the same time period showing an overall increasing trend in air temperature at each airport site.

Page 8 line 244 – this sentence does not seem to make sense, I think you actually mean something like “Monitoring thunderstorms and related atmospheric phenomena play a very important role in supervising the movement of aircraft.”

Page 8 line 247 – what do you mean by “fault”?

Page 8 line 254 – what do you mean by “procedures describing flying”

Page 9 paragraph beginning line 268 – same as comment for page 8 line 239 above, please include some evidence of this.  It is a very important point, that storm intensity is increasing, which impacts aircraft operations.  Please include the relevant data so the importance cannot be refuted!

Page 9 line 278 – how are these catastrophe phenomena defined?  For example rainfall, low visibility and wind shear could all occur together as part of one storm, is that storm then counted 3 times?  Also where are these data for?  Poland, the whole world, one airline, one airport?

Figures 1-3 – is there any particular reason for the two sites included on each plot?  If not, it might be better to either show them all on one plot, or individually.

Figures 6 and 7 – captions are not in English

Author Response

Dear Reviewer,

Yours sincerely,

Jaroslaw KOZUBA, prof. SUT

Round 2

Reviewer 2 Report

The paper has been greatly improved - thank you.
There are still some minor English editing changes required, and the caption to Figures 6 and 7 are still not in English.

Author Response

Dear Reviewer,

Yours Sincerely,

Jarosław KOZUBA, prof. SUT
